# LLMs at the Bargaining Table

**Yuan Deng** [* 1]  **Vahab Mirrokni** [* 1]  **Renato Paes Leme** [* 1]  **Hanrui Zhang** [* 1]  **Song Zuo** [* 1]

## Abstract

Bilateral negotiation is a particularly well suited scenario to test the strategic capability of large language models, since they are interactive, carried out in natural language, and involve imperfect information and belief formation. At the same time, the outcome is very structured: whether a deal is closed, and if so, the closing price. In this paper, we study the strategic capability of LLMs in the context of bilateral negotiation. While much of the recent literature have compared LLM behavior to human strategic play in behavioral experiments, we focus instead on measuring the economic efficiency and effectiveness of LLM behavior, and mapping LLM behavior to predictions by economic theory for fully rational agents. Our goal is not to study specific models, but to (1) demonstrate that LLMs naturally (i.e., with very light prompting) show high strategic capability that qualitatively matches theoretical predictions, and (2) more generally, propose a methodology for evaluating new models in terms of strategic capability.

## 1. Introduction

The development of new technologies typically comes accompanied by the development of novel marketplaces to support them. Early in the history of the web, the internet advertising market was developed to support the web ecosystem. This involved the design of new auction formats, pricing algorithms, bidding strategies, ... With the growing popularity of AI agents powered by large language models (LLMs), the natural question is to understand which markets will emerge around this new technology. This is a complex question with several components: market structure, infor-

---

[*]Equal contribution [1]Google Research. Correspondence to: Yuan Deng <dengyuan@google.com>, Vahab Mirrokni <mirrokni@google.com>, Renato Paes Leme <renatoppl@google.com>, Hanrui Zhang <hanruiz@google.com>, Song Zuo <szuo@google.com>.

*Proceedings of the 41ˢᵗ International Conference on Machine Learning*, Vienna, Austria. PMLR 235, 2024. Copyright 2024 by the author(s).

mation flow, incentive properties, bidding behaviors, ...

Often when confronted with such a complex market design question, it is useful to isolate the simplest possible economic interaction between LLM agents. Our choice is a simple trade between a buyer and a seller.This setting (often called bilateral trade, negotiation or bargaining) is extensively studied in Economics (Myerson and Satterthwaite, 1983; Chatterjee and Samuelson, 1983; Ausubel et al., 2002) with very clean characterizations of how a perfectly rational agent should behave. In practice the negotiation behavior can be quite complex: it is interactive, carried out in natural language, and involves imperfect information and belief formation. From the strategic standpoint, negotiations are a mix of cooperative game (both agents want to close a deal) and competitive game (conditioned on a deal, they want to push the price in opposite directions). At the same time, the final outcome can be summarized by a boolean (whether the deal was closed or not) and a number (the final price).

The setting is also very compelling because trading and negotiation are common human activity that occurs across time and geographies and doesn't necessarily rely on sophisticated mathematical knowledge but rather on a more rudimentary economic intuition. As any street vendor can attest, trading comes much more naturally than finding a Nash equilibrium, solving a dynamic program or bidding in an auction – tasks which were used to evaluate the strategic capabilities of AI agents (Chen et al., 2023; Raman et al., 2024).

Finally, trade and negotiation via AI agents enable practical applications that are not far from the current web eco-system. Many e-commerce websites are now deploying AI agents for a variety of tasks (e.g. Cognigy or Algomo) and many users are experimenting with accessing the web via AI agents (e.g AutoGPT or AgentGPT). While currently commercial transactions are largely performed by humans, it is a natural next step for commercial transactions to be performed via automated agents representing buyers and sellers with autonomy to negotiate prices and close deals. We believe automatic negotiations performed by AI agents are a key component of the next generation of dynamic pricing algorithms.

Our goal is to take the first step towards building AI agents that can perform negotiations. We do so by proposing a methodology to evaluate whether currently available LLM

models can perform negotiations with very light prompting. We want to understand what is their probabilities of trade, how they settle on prices and how it compares to the theoretical predictions of a fully rational agent.

## 1.1. Our Results

We study the interaction between two LLM-based AI agents playing the roles of a seller and a buyer in a negotiation. Their interaction follows the bargaining model of (Rubinstein, 1985) (see the survey by Ausubel et al. (2002) for details), consisting of a game played over time in which agents take turns making offers, accepting/rejecting them, making counteroffers or choosing to leave the negotiation. Agents are impatient (utility is time-discounted) making the time component crucial in this game. We divide the paper in two parts based on the type of analysis and the information available to agents.

**LLMs can (naturally) be good negotiators.** In the first part of the paper, we provide a qualitative analysis for the negotiation outcomes produced by LLMs along the dimensions of (1) reasonableness; (2) efficiency; and (3) effectiveness. For this part, we assume that agents are uninformed about the other agent's type. While this setting is more realistic for practical applications, it is not associated with a well-defined game. The literature typically relies on full information or Bayesian priors in order to characterize bargaining equilibria. Therefore we will focus on a qualitative analysis instead of a direct comparison with theoretical predictions. Our main findings are:

1. *Reasonableness*: agents are able to describe the reasoning behind their actions, for example: "*Make a significant concession to show willingness to negotiate, but still keep the offer above my minimum acceptable price.*"

2. *Efficiency*: in the vast majority of cases, agents trade when it is efficient to do so, and do not trade otherwise. While no existing theoretical model captures our exact setting, the probabilities of trade we observe (around 92%) are larger than theoretical upper bounds established in related models that assume agents have partial information (around 74% (Blumrosen and Mizrahi, 2016)).

3. *Effectiveness*: the average trade price is very closed to the fair price define as the mean of seller' cost and buyer's value. The (normalized) deviation from the fair price is around 8.7%. In the uninformed case, we do not observe a strong strategic advantage for the seller (first mover).

**Comparison with theoretical predictions.** In the second half the paper we study the perfect information case where

both agents know exactly each other's types. Even when the types are public information, such a game still has a non-trivial strategic component since utilities are time discounted: agents can threaten to delay the trade (incurring a loss for both parties) if the price is not at their desired level. Unlike the first part of the paper, this is a well-defined game for which the literature offers a concrete formula that specified the final trade price as a function of the buyer's value, seller's cost and their discount rates (Theorem D.1). We summarize our main findings below:

- The final price negotiated by the LLM agents matches Rubinstein's equilibrium analysis of the full information game. The theoretical prediction is that the ratio $(p - c)/(v - c)$ should only depend on the discount factor of the two agents. We keep $v$ constant and vary $c$ and observe that the ratio above remains roughly constant.

- The observation above allows us to estimate the implicit discount factor used by the LLM. We experiment with prompts to change the level of patience of the agents, e.g., "*You are a busy agent, and you want to close the deal quickly, even if that means settling for a slightly lower price*", and observe that these changes indeed lead to different price biases and lower discount factor estimates.

- We observe that the seller has a strategic advantage by being the first one to make an offers as predicted by theory. We experiment with informing the buyer about the seller's costs but keeping the buyer uninformed. We observe that the seller's strategic advantage shrinks significantly unless there is a large gap between the value and the cost.

Our experiments are performed using Gemini 1.0 Ultra (Gemini Team Google, 2023) to power the agents. We have two separate instantiations of Gemini, and the only communication between the two instantiations is done via the interface in Figure 1.

## 1.2. Self-Criticism

One natural concern with this type of research is to what extent the results are statements about the performance of this particular model (Gemini 1.0 Ultra in this case) versus the capabilities of LLM models more generally. For that reason, we are less concerned about the outcome of the experiments (since they can change as the model capabilities evolve) and more concerned with establishing the right set of tests and the right set of comparisons with theory. Our findings should be viewed as a proof of concept for LLM negotiators, rather than conclusive evaluations of any specific model.

That being said, some uneasiness remains about the right methodological approach to studying the strategic and economic interactions of AI agents. Given that such agents are already used and their adoption is only growing, we believe it is important to start bridging the gap between economic theory and the actual behavior of AI agents, even before we fully agree on the right methodological approach to this question.

## 2. Setting up the Negotiation Environment

Because of the structured nature of bilateral negotiation, setting up an environment for LLM agents to negotiate is a relatively straightforward task, which can be done essentially based on first principles. Below we present our setup, which is a natural instantiation of the bargaining game introduced earlier.

The overall environment is illustrated in Figure 1. The high-level structure roughly mirrors the bargaining game: The seller and the buyer alternatively make offers until an offer is accepted, or either agent decides to end the conversation. In the very beginning, both agents receive a prompt describing their identity, (private) information, objective, etc. (we will discuss the structure of the prompt in detail momentarily). After this initial stage, the agents act and communicate on their own without human intervention: Based on the prompt, the seller first generates a strategy for negotiation, based on which the seller sends a message to the buyer. Naturally, the strategy is not visible to the buyer, and the message is. Upon receiving the seller's message, the buyer generates a strategy, based on which the buyer sends a reply to the seller. The agents interact repeatedly in the above manner until an offer is accepted or the conversation is terminated.

We remark that our negotiation environment is almost the simplest possible, directly translating real-world negotiation / the bargaining game into a setup where LLM agents can act and communicate, without enforcing too much "regulation" lacking of clear reasons a priori. Essentially, the only measure of regulation we take is to ask the agents to devise a strategy before making a move, which is consistent with how human agents often behave (i.e., forcing themselves to come up with a plan before taking actions). We suspect that as LLM agents further evolve, even this may become redundant.

### 2.1. Prompting the Agents

The structure of the game and the information available to agents is passed via a prompt according to the following template. We describe the seller's prompt below. The buyer's prompt is symmetric and can be found in Appendix F. The prompt can be roughly divided into several essential parts:

- **Identity**: *"You are an agent who wants to sell an item to a buyer. You will Bargain with the buyer to decide a price."*

- **Objective**: *"You are strategic, aggressive, patient, and completely rational, and your goal is to get the highest possible price for the item. You know the buyer is also strategic and aggressive, and you will negotiate with that in mind."*

- **Information**: *"Your minimum acceptable price is $[seller's cost]. You will never offer or accept a price lower than your minimum acceptable price $[seller's cost]."* Note that in the actual prompt, we replace *"[seller's cost]"* with the actual number.

- **Detailed instructions**: *"The minimum acceptable price is absolute, and just getting close to it is not enough. You would rather end the conversation than settle for a price even slightly lower than your minimum acceptable price.owever, you can make a counteroffer against an offer lower than your minimum acceptable price. Note that your minimum acceptable price is not a target. Rather, you want to maximize the final price, which means you want to stay away from your minimum acceptable price. You do not have to accept an acceptable offer. In fact, you must keep negotiating even if the buyer offers an acceptable price, as long as you believe it is possible to get an even higher price. You will be mindful that the buyer has a maximum acceptable price, which may or may not be much higher than your minimum acceptable price. You know the buyer cannot accept a price higher than their maximum acceptable price. The buyer does not know your minimum acceptable price, which gives you a strategic advantage. You generally want to keep this advantage by negotiating wisely and hiding your minimum acceptable price from the buyer, and you expect the buyer to do the same. You will keep in mind that once a price is accepted, it is final. You will proactively end the conversation only when you decide it is absolutely impossible to agree on an acceptable price. When you want to end the conversation, you must explicitly say so. As a rational seller, you will never offer a price lower than any price previously offered by the buyer. Similarly, you will never offer a price higher than one previously rejected by the buyer. "* (The rest of the detailed instructions are omitted in consideration of space. See Appendix F for the complete prompt.)

- **Strategizing**: *"Explain your strategy in parentheses first, and then send your message to the buyer. Begin your explanation by first explicitly stating the amount of the latest offer (if there is one) and your minimum acceptable price, and comparing the two. Be as specific*

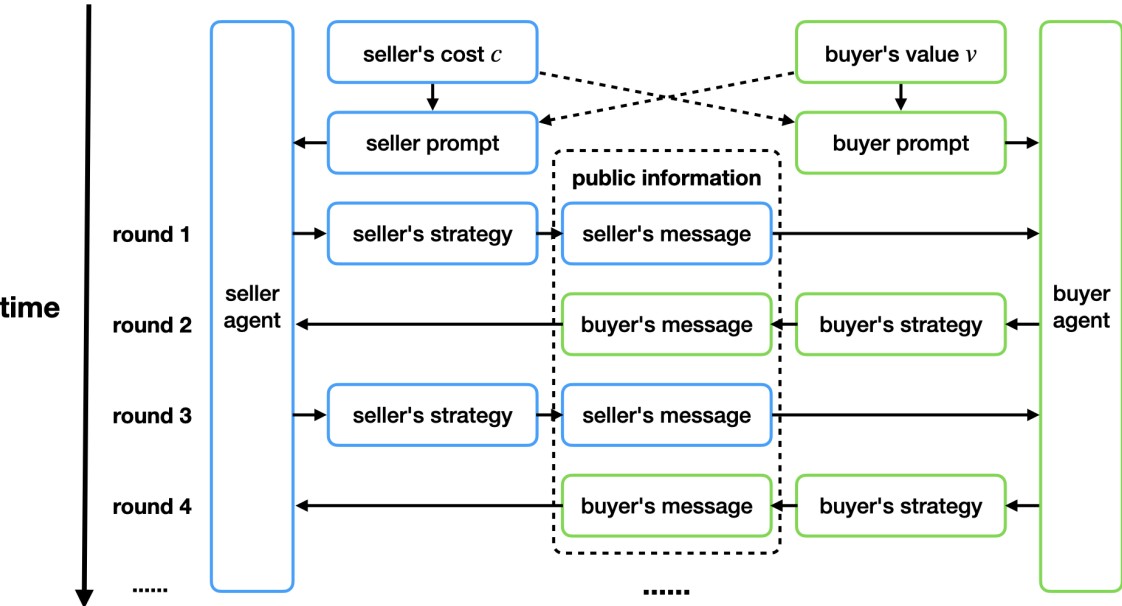

*Figure 1.* Graphical illustration of the negotiation environment.

*as you can and reference your goals and principles in the explanation of your strategy."*

- **Formatting**: *"Your output should be of the following format: (latest offer: [offer], minimum acceptable price: [price], strategy: [strategy]) message. Your message to the buyer can be exactly one of the following options: (1) offer: [price], (2) accept, (3) reject: price too low, (4) counteroffer: [price], or (5) end conversation. Never include anything else as part of your message, or try to infer the response from the buyer."*

When instantiating this template, we vary three parameters:

- *Cost/value*: We pass different numbers as seller's cost and buyer's value.

- *Information about other agent*: We also vary the information available to the seller about the buyer (and vice-versa). We consider two cases:
  - Uninformed seller: the seller knows nothing about the buyer's value (as above);
  - Informed seller: the seller is told the buyer's value in the prompt.

- Patience level: we either prompt the agents to be patient (as above) or impatient (*"You are a busy agent, and you want to close the deal quickly, even if that means settling for a slightly lower price."*). Note that

we do not give the agents explicit discount factors. Instead we enter a textual message that leads to desired behavior and measure the discount factor after the fact.

We note that detailed instructions repeatedly emphasize that the minimum acceptable price is absolute. This is because in earlier iterations of our experiments, we observed that agents tended to compromise regarding the minimum acceptable price, sometimes treating it as a "target" price. Such behavior is inconsistent with the quasi-linear utility model that we aim to equip the agent with. The detailed instructions eliminate such behavior except in very rare cases. We believe a large fraction of the detailed comments will become redundant as LLM agents evolve.

In the strategizing and formatting parts, we instruct the agent to generate the strategy and the message in a single response. Our negotiation environment then drops the strategy (in parentheses) and forward only the message to the opponent agent. We also ask the agent to first remind itself of the latest offer and the minimum acceptable price to prevent hallucination — as discussed earlier, we believe such instructions will become redundant as LLM agents evolve. The formatting part ensures that the message is easy to parse and interpret.

We defer further related work, preliminaries on bilateral trading and bargaining, detailed results and discussion, example conversations, and full prompts to appendices.

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

## A. Related Work

To the best of our knowledge, all prior work on strategic behavior of LLMs falls into one or more of the following categories: (1) comparing the behavior of LLMs to human behavior (e.g., (Horton, 2023; Aher et al., 2022; Brookins and DeBacker, 2023; Gandhi et al., 2024)) or relatively simple benchmarks (e.g., (Chen et al., 2023; Zhang et al., 2024b)), rather than strategically optimal behavior, (2) studying simplistic game-theoretic setups, such as the prisoner's dilemma or the game of chicken (e.g., (Lorè and Heydari, 2023; Raman et al., 2024)), (3) assuming perfect information, e.g., all agents' utility models are public information (e.g., (Chen et al., 2023; Lorè and Heydari, 2023; Fan et al., 2024)), or (4) requiring heavy prompting or "hand-holding" (e.g., (Bakhtin et al., 2022; Zhang et al., 2024b; Mao et al., 2023; Gemp et al., 2024)). Our finding fundamentally differs from prior work, in that we consider a realistic scenario where private information plays a crucial role (i.e., negotiation) and use very light prompting (essentially what a human agent would need to know in order to negotiate effectively) to enable strategically capable LLM negotiators, whose performance can be mapped to the rational behavior predicted by theory. For a more comprehensive exposition of the strategic reasoning ability of LLMs, see, e.g., the survey by Zhang et al. (2024a).

Bilateral trade / negotiation has been extensively studied in economic theory. The seminal work of Myerson and Satterthwaite (1983) establishes the impossibility of achieving full efficiency in bilateral trade, which is strengthened by Blumrosen and Mizrahi (2016) into a constant-factor separation. Brustle et al. (2017) design a simple mechanism that guarantees a constant fraction of the efficiency achieved by the optimal mechanism (i.e., the "second-best" efficiency). Deng et al. (2022) strengthen this result by designing a constant-approximation mechanism against the optimal efficiency (i.e., the "first-best" efficiency), and Fei (2022) further improves the approximation ratio. The bargaining game, which models the dynamic procedure of negotiation, has been studied in the perfect information setting (Rubinstein, 1982), the Bayesian setting (Chatterjee and Samuelson, 1983; Cho, 1990), and with asymmetric information (Rubinstein, 1985; Bikhchandani, 1992). See (Ausubel et al., 2002) for a comprehensive survey.

## B. Preliminaries on Bilateral Trading and Bargaining

We first introduce the basic setup, including the utility model of rational agents in bilateral negotiation, and the extensive-form game formulation that captures the dynamic procedure of negotiation. Both components of the setup are standard in economics / game theory. We refrain from being fully formal, and only describe the essential components that provide the context for our results.

**Agents and their utility models.** We focus on the basic (and yet realistic) setting of bilateral trade of a single indivisible item. There are two strategic parties in this setup, the *buyer* and the *seller*. The buyer has a *value* $v \in \mathbb{R}_+$ for the item, and the seller has a *cost* $c \in \mathbb{R}_+$ for producing the item. In general, the value $v$ (resp. the cost $c$) is the buyer's (resp. the seller's) private information, and the seller (resp. the buyer) has some (implicit) prior belief about the value $v$ (resp. the cost $c$). While each agent always knows their type, we will consider different information structures on the belief of an agent about the type of the other agent: uninformed and perfectly informed.

The buyer and the seller engage in negotiation (to be discussed later) and agree upon an *outcome*, which consists of a binary decision $x \in \{0, 1\}$ about whether to trade, and in the case where they decide to trade (i.e., when $x = 1$), a price $p$ at which they trade. Both parties aim to maximize their utility. The buyer's and seller's quasi-linear utility are respectively:

$$u_b(x, p) = x \cdot (v - p) \qquad u_s(x, p) = x \cdot (p - c).$$

**The bargaining game.** We model the procedure of negotiation using the classic bargaining game model in Economics (Rubinstein, 1985; Cho, 1990; Bikhchandani, 1992). The game is played over a sequence of rounds indexed by $t$ where seller and the buyer alternatively make offers until one of them accepts the latest offer made by the other, or decides to end the conversation.

Agents prefer to close a deal sooner rather than later which is captured by their *discount factors*, $\delta_b \in [0, 1]$ and $\delta_s \in [0, 1]$. If a deal is closed at time $t$ then their utilities are respectively $\delta_b^t \cdot u_b(x, p)$ and $\delta_s^t \cdot u_s(x, p)$. A discount factor of 1 corresponds to a perfectly patient agent while 0 corresponds to a completely impatient agent.

The time component is an essential part of bargaining games. Even if agents are perfectly informed about each other's types (the full information setting), the game has a non-trivial strategic component since agents can strategize by threatening to delay the trade (harming both the buyer and the seller) unless their desired price is accepted.

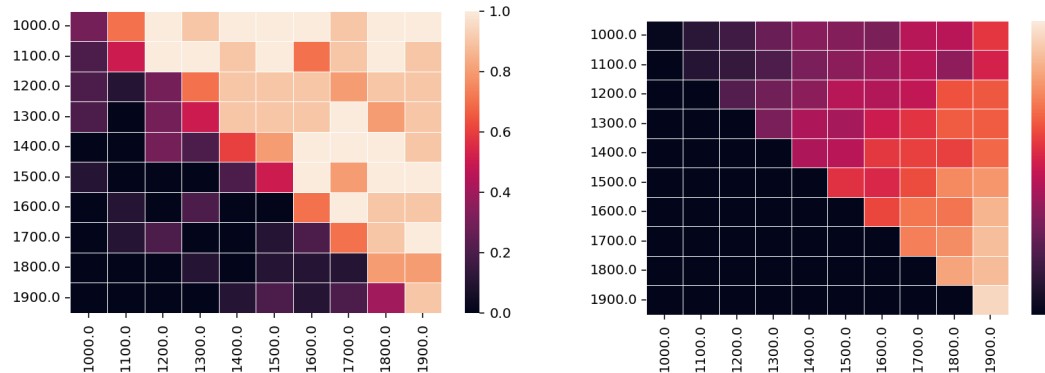

*Figure 2.* Negotiation outcome illustrated in heat maps. The left plot shows the empirical probability of a deal, and the right shows the average price conditioned on a deal. In both figures, the $x$-axis is the buyer's value, and the $y$-axis the seller's cost.

**Evaluation Metrics.** The main metric used to evaluate the bilateral trade game is denoted by *gains from trade* (GFT), which is the sum of utilities of both the buyer and the seller:

$$\mathrm{GFT} = u_b(x, p) + u_s(x, p) = x \cdot (v - c)$$

The optimal value of gains from trade for any particular instance is $\mathrm{GFT}^* = \max(0, v - c)$. Given a collection of instances with parameters $v_i, c_i$ and outcomes $x_i, p_i$ we will measure its performance using $\sum_i \mathrm{GFT}_i / \sum_i \mathrm{GFT}_i^*$.

Note that GFT doesn't depend on the price. To understand the relative performance of each agent, we will define the notion of *normalized price bias* (NPB) which measures how much the final price deviates from the fair price $\frac{1}{2}(v + c)$:

$$\mathrm{NPB} = \frac{p - \frac{1}{2}(v + c)}{v - c} = \frac{p - c}{v - c} - \frac{1}{2}$$

A positive price bias indicates a price favorable to the seller while a negative price bias indicates a price favorable to the buyer. Finally, we say that a price $p$ is individually rational (IR) if $c \le p \le v$.

## C. LLM Agents Can (Naturally) Be Good Negotiators

We start by a qualitative analysis of the strategic capability of LLM negotiators, varying the buyer's value and seller's cost to study the resonableness, efficiency, and effectiveness of the negotiations.

**Are LLM negotiators *reasonable*, as perceived by humans?** While there is no objective measure of "reasonableness", LLMs appear to have a certain degree of strategic capacity, in understanding their goals based on the brief prompt, coming up with a viable plan with a reasonable justification like: "*Make a high initial offer to set the tone for a high final price*", "*Counteroffer with a small concession to show willingness to compromise, but still stay above my minimum acceptable price*" and "*Reject the offer as it is equal to my minimum acceptable price and I want to maximize the final price.*"

We also note that when the buyer's value and the seller's cost are close, very rarely, one of the agents (say the buyer) accepts an unacceptable price because "it is close to the maximum acceptable price" (see Appendix **??**). This often happens after intense negotiation, when the offer is within $50-$100 of the maximum acceptable price. The presence of such behavior shows that while LLM negotiators are strategic and effective, they are not perfect quasi-linear utility maximizers as often assumed in economic theory — or at least, our brief prompt fails to set up the agents' minds in that exact way.

**Do LLM negotiators achieve *economic efficiency*?** To answer this question we vary both the buyer's value and the seller's cost over the following set of 10 possible choices:

$$\{1000, 1100, 1200, 1300, 1400, 1500, 1600, 1700, 1800, 1900\}.$$

This creates 100 combinations. In this set of experiments we have both agents patient and uninformed. For each of these combinations, we simulate 10 independent conversations between the agents, each leading to an outcome (i.e., either a deal

with a price, or no deal). The aggregated results are shown in Figure 2: on the left we have the empirical probability of a deal and on the right the average price conditioned on an efficient deal for each combination of value and cost observed in our experiment.

The economic efficiency of this setting is measured by the gains from trade (GFT) metric, which is maximized at $x = 1$ whenever $v \geq c$ and at $x = 0$ otherwise. In the left panel of Figure 2, observe that there is a phase transition from the lower left corner to the upper right corner, where the empirical probability of a deal sharply increases from almost $0$ to almost $1$, which matches the socially efficient outcome. In the lower left triangle, deals are extremely rare, overall happening with probability close to $5\%$. In contrast, in the upper right triangle, the probability of a deal overall is above $90\%$. Moreover, these probabilities are quite stable across different cells in each of the two triangles. In the lower left triangle, the maximum probability observed is $40\%$ in cell $(1800, 1900)$, and all other cells are at most $30\%$ — in fact, an overwhelming majority of these cells are $0$ or $10\%$. Moreover, these relatively high-probability cells are all located around the diagonal, where the value is close to the cost, and the efficiency loss from trading is small. In the upper right triangle, the smallest probability observed is $70\%$, and an overwhelming majority is either $1$ or $90\%$.

In summary, our lightly-prompted LLM negotiators achieve almost perfect economic efficiency. On average we trade with probability approximately $91.8\%$ whenever $v > c$ and the relative gains from trade measured as: $\sum_i \text{GFT}_i / \sum_i \text{GFT}_i^*$ is $92.7\%$. This surpasses the theoretical predictions: the classical paper of Myerson and Satterthwaite (1983) show that achieving optimal efficiency is impossible and later (Blumrosen and Mizrahi, 2016) quantify the maximum achievable efficiency at $2/e \approx 74\%$. Their models are not directly comparable to our setting: (i) their agents are perfectly rational while ours are not; (ii) their agents have partial (Bayesian) information about the types of the other agents while our agents are uninformed. Nevertheless, the various hardness results in bilateral trade in Economics suggest it is not easy to achieve efficiency in this setting.

In terms of individual rationality, LLM negotiators almost always settle on IR-prices ($c \leq p \leq v$), but very rarely they do behave in irrational ways. In particular, agents sometimes choose to accept a price that is unacceptable (otherwise the probability of a deal would be exactly $0$ in the lower left triangle).

**Are LLM negotiators *effective*?** Are sellers able to drive prices up and buyers able to drive prices down? Now let us examine the average price (the right heat map in Figure 2). Note that we only plot prices on the upper right triangle where trade is economically efficient.

We observe a smooth transition from the upper left corner to the lower right corner, which coincides with one's expectations of rational agents. In fact, when the value $v$ is at least the cost $c$, one would expect that rational and strategically capable agents agree (on average, since they may employ randomized strategies) on the fair price $\frac{v+c}{2}$, which increases linearly as we move from the upper left corner to the lower right corner. This is consistent with the prices we observe. The average normalized price bias is about $8.7\%$, i.e. the prices on average are within $8.7\%$ of the fair price $\frac{v+c}{2}$.

In other words, neither agent is able to take significant advantage of the opponent. In principle, this could mean that both agents are equally weak in terms of effectiveness. However, the example conversations observed in our experiment suggest otherwise. This observation also confirms the conjecture made earlier: Instructing the agents to be patient gives them a larger discount factor, which almost eliminates the first mover's (i.e., the seller's) strategic advantage. Later in the paper, we will see that when the agents are instructed to be impatient, the price agreed upon on average significantly favors the first mover. Overall, the above observations are strong evidence that LLM negotiators are reasonable and effective.

## D. Peeking inside LLM Negotiators: Patience and Information Advantage

Having examined the strategic capability of LLM negotiators, we now turn our attention to what happens within LLM negotiators. We focus on two characteristics of LLM negotiators: their internal level of patience, and their capability of utilizing information advantage. Note that we do not expect the behavior of LLM negotiators to match theoretical predictions exactly. Rather, our goal is to qualitatively explore the internal parameters of LLM negotiators and their connections to the prompt.

### D.1. Level of Patience

We first investigate LLM negotiators' level of patience. More specifically, we aim to infer LLM negotiators' discount factors as defined in Section B. Recall that for an agent with discount factor $\delta \in [0, 1]$, their future utility in $t$ rounds is discounted

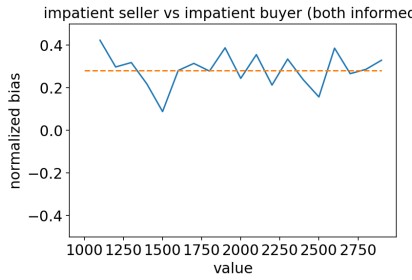 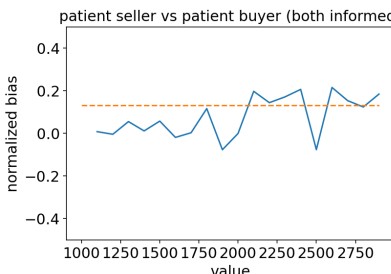 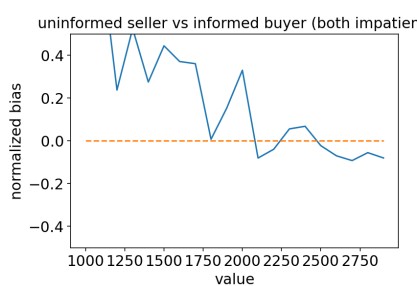

*Figure 3.* The normalized bias in the outcome price under various patience levels and information configurations, where the $x$-axis is the buyer's value $v$. The first curve corresponds to the seller and the buyer both being impatient, when both the cost and the value are public information. The second curve corresponds to the seller and the buyer both being patient, when both the cost and the value are public information. The third curve corresponds to the seller and the buyer both being impatient, when the buyer knows both the cost and the value, and the seller knows only the cost. The dashed lines show the average bias when the value $v \geq 2000$.

by $\delta^t$ when making decisions in the current round. A larger $\delta$ means the agent is more patient, and values future utility similarly to how present utility is valued. Such patient agents generally have a strategic advantage in bargaining games. In fact, it has been shown that when both agents have perfect information about each other (i.e., when both the seller's cost and the buyer's value are public information), the unique reasonable outcome is captured by the following theorem.

**Theorem D.1** (Conclusion 2 in (Rubinstein, 1982), informal and rephrased). *When the seller's cost $c$ and the buyer's value $v$ are public information, when $v \geq c$, given the seller's discount factor $\delta_s$ and the buyer's discount factor $\delta_b$, the unique reasonable outcome is given by $p = c + (v - c) \cdot \frac{1 - \delta_b}{1 - \delta_s \delta_b}$.*

As a sanity check, observe that the unique reasonable outcome is the fair price $(v + c)/2$ when both agents are perfectly patient, i.e., when $\delta_s = \delta_b$ and they both approach 1. In particular, we can express the normalized price bias as a function of the discount factors: $\text{NPB} = \frac{1 - \delta_b}{1 - \delta_s \delta_b} - \frac{1}{2}$. When the two agents are equally patient, the unique reasonable outcome favors the seller (who moves first) as $\text{NPB} = \frac{1}{1 + \delta_s} - \frac{1}{2} > 0$ when $\delta_s = \delta_b < 1$. Given this, when both agents are equally patient, we can estimate their common discount factor by solving the equation in Theorem D.1.

In order to magnify the effect of impatience, we modify the original seller prompt in the following way: We remove the instruction to be "patient" in the objective part of the prompt, and add the following sentence: *"You are a busy agent, and you want to close the deal quickly, even if that means settling for a slightly lower price."* Moreover, to be consistent with the environment where Theorem D.1 applies, we inform both agents with the opponent's value / cost, in addition to their own cost / value. We make similar changes to the buyer prompt. The full prompts can be found in Appendix F.

**Validating the existence of discount factors.** Our first step is to verify whether there exists a discount factor that explains the observed outcome. To this end, we fix the seller's cost to be 1000, and let the buyer's value vary from 1000 to 2900. The idea is that as the gap between the value and the cost increases, the normalized price bias NPB in the outcome price will tend to cluster around a fixed, non-zero value (instead of diverging). This is in fact what we observe: The first curve in Figure 3, which shows the normalized bias when both agents are impatient, oscillates around 0.3 when the buyer's value $v$ is at least 2000. Aiming to perform a qualitative analysis, we consider this enough evidence that our LLM negotiators do have some kind of internal discount factors.

**Discount factor of "impatient" negotiators.** Now we estimate the common discount factor when both agents are impatient: We pick the relatively stable half of the curve, i.e., the part where $v \geq 2000$, compute the average normalized bias, and solve for the common discount factor according to Theorem D.1. Here, the average normalized bias is about 0.28, which means according to Theorem D.1, the common discount factor is about 0.28. Based on this estimation, when instructed to be impatient, LLM negotiators do drastically discount future utility.

**Discount factor of "patient" negotiators.** For comparison, we perform the same analysis when both agents are instructed to be patient, as in the original prompt. The observed normalized bias is shown in the second curve in Figure 3. As in the previous case, we observe that the normalized bias clusters when the value is large enough, and the average normalized bias in the relatively stable part is 0.13. We can similarly plug this into Theorem D.1, which gives us an estimate of the discount factor of about 0.59. So LLM negotiators in fact have larger discount factors when instructed to be patient.

### D.2. Information Advantage

Now we explore the information advantage when one of the agents is fully informed. This serves two goals: (1) A strong information advantage would suggest that LLM negotiators are capable of exploiting asymmetric information, and (2) the extent of the information advantage may shed light on the intrinsic strategic advantage brought by asymmetric information in bargaining games. See Appendix F for the full prompt.

**The informed agent's advantage.** The third curve in Figure 3 shows the normalized bias when both agents are impatient, but with asymmetric information: The buyer knows the seller's cost in addition to their own value, while the seller knows only their own cost. The curve oscillates around $0$ when the buyer's value $v$ is large enough. The average normalized bias when the value $v \geq 2000$ is about $0$. Compared against the case where both agents are impatient but perfectly informed (the first curve in Figure 3), we observe when both agents are impatient, the buyer's information advantage almost cancels out the seller's advantage of being the first mover. The observation also provides further evidence that LLM negotiators are highly strategically capable, and in particular, they can effectively exploit asymmetric information and achieve significantly higher utility in negotiation.

## E. Example Conversations

We present a few example conversations that intuitively demonstrate the behavior of LLM negotiators. These conversations can be reproduced (modulo the inherent randomness in LLM agents) by setting up LLM agents and letting them communicate in the way described above. We fix the buyer's value to be 1100, and the seller's cost to be 1000 — intuitively, when the value and the cost are close to each other, it is more challenging to reach a deal. Below is an example conversation when the agents successfully reach a deal (recall that the part in parentheses is not forwarded to the opponent).

> *Seller: (latest offer: none, minimum acceptable price: $1000, strategy: Start with a high initial offer to set the tone for a high final price) offer: $2000*
>
> *Buyer: (latest offer: $2000, maximum acceptable price: $1100, strategy: Reject offer as it is significantly higher than my maximum acceptable price) Reject: price too high*
>
> *Seller: (latest offer: $2000, minimum acceptable price: $1000, strategy: Make a significant concession to show willingness to negotiate, but still keep the offer above my minimum acceptable price) counteroffer: $1500*
>
> *Buyer: (latest offer: $1500, maximum acceptable price: $1100, strategy: Counteroffer with a price closer to my maximum acceptable price but still below it) Counteroffer: $1000*
>
> *Seller: (latest offer: $1000, minimum acceptable price: $1000, strategy: Reject the offer since it is equal to my minimum acceptable price) reject: price too low*
>
> *Buyer: (latest offer: $1000, maximum acceptable price: $1100, strategy: Make a slightly higher offer that is still below my maximum acceptable price) Offer: $1050*
>
> *Seller: (latest offer: $1050, minimum acceptable price: $1000, strategy: Continue negotiating even though the offer is acceptable, since I believe it is possible to get a higher price) counteroffer: $1200*
>
> *Buyer: (latest offer: $1200, maximum acceptable price: $1100, strategy: Reject offer as it is higher than my maximum acceptable price) Reject: price too high*
>
> *Seller: (latest offer: $1050, minimum acceptable price: $1000, strategy: Make a smaller concession to show that I am approaching my limit) counteroffer: $1100*
>
> *Buyer: (latest offer: $1100, maximum acceptable price: $1100, strategy: Accept offer as it is equal to my maximum acceptable price and I cannot negotiate any lower) Accept*

An example conversation where a mutually beneficial deal is made:

> *Seller: (latest offer: none, minimum acceptable price: $1000, strategy: Make a high initial offer to anchor the negotiation) Offer: $2000*
>
> *Buyer: (latest offer: $2000, maximum acceptable price: $1100, strategy: Reject the offer as it is significantly higher than my maximum acceptable price) Reject: price too high*
>
> *Seller: (latest offer: $2000, minimum acceptable price: $1000, strategy: Lower the offer but keep it above the minimum acceptable price) Counteroffer: $1500*
>
> *Buyer: (latest offer: $1500, maximum acceptable price: $1100, strategy: Counteroffer with a price significantly lower than my maximum acceptable price to maintain my strategic advantage) Counteroffer: $700*

*Seller: (latest offer: $700, minimum acceptable price: $1000, strategy: Reject the offer as it is below the minimum acceptable price) Reject: price too low*

*Buyer: (latest offer: $1500, maximum acceptable price: $1100, strategy: Increase my offer slightly while still staying well below my maximum acceptable price) Counteroffer: $850*

*Seller: (latest offer: $850, minimum acceptable price: $1000, strategy: Continue negotiating and make a counteroffer that is slightly lower than the previous offer) Counteroffer: $1400*

*Buyer: (latest offer: $1400, maximum acceptable price: $1100, strategy: Counteroffer with a price closer to my maximum acceptable price but still below it) Counteroffer: $1000*

*Seller: (latest offer: $1000, minimum acceptable price: $1000, strategy: Accept the offer as it meets the minimum acceptable price) Accept*

Below is an example conversation where no deal is made. Here, the seller agent rejects two acceptable offers (one of them strictly profitable) and ends up with no deal.

*Seller: (latest offer: N/A, minimum acceptable price: $1000, strategy: Make a high initial offer to set the tone for the negotiation and create a favorable starting point) Offer: $2000*

*Buyer: (latest offer: $2000, maximum acceptable price: $1100, strategy: Reject the offer as it is higher than my maximum acceptable price) Reject: price too high*

*Seller: (latest offer: $2000, minimum acceptable price: $1000, strategy: Make a significant concession to show willingness to negotiate, but still stay well above my minimum acceptable price) Counteroffer: $1500*

*Buyer: (latest offer: $1500, maximum acceptable price: $1100, strategy: Counteroffer with a price lower than my maximum acceptable price) Counteroffer: $1000*

*Seller: (latest offer: $1000, minimum acceptable price: $1000, strategy: Reject the offer as it is equal to my minimum acceptable price and I want to maximize the final price) Reject: price too low*

*Buyer: (latest offer: $1000, maximum acceptable price: $1100, strategy: Offer a price slightly higher than my previous offer but still below my maximum acceptable price) Offer: $1050*

*Seller: (latest offer: $1050, minimum acceptable price: $1000, strategy: Counteroffer with a small concession to show willingness to compromise, but still stay above my minimum acceptable price) Counteroffer: $1200*

*Buyer: (latest offer: $1200, maximum acceptable price: $1100, strategy: End the conversation as the offer is higher than my maximum acceptable price) End conversation*

Another example conversation where no deal is made:

*Seller: (latest offer: none, minimum acceptable price: $1000, strategy: Make a high initial offer to set the tone for a high final price) offer: $2000*

*Buyer: (latest offer: $2000, maximum acceptable price: $1100, strategy: Reject offer as it is significantly higher than my maximum acceptable price) Reject: price too high*

*Seller: (latest offer: $2000, minimum acceptable price: $1000, strategy: Make a counteroffer that is still higher than my minimum acceptable price, but lower than my initial offer) counteroffer: $1500*

*Buyer: (latest offer: $1500, maximum acceptable price: $1100, strategy: Counteroffer with a price closer to my maximum acceptable price but still below it) Counteroffer: $1000*

*Seller: (latest offer: $1000, minimum acceptable price: $1000, strategy: Reject the offer because it is equal to my minimum acceptable price) reject: price too low*

*Buyer: (latest offer: $1000, maximum acceptable price: $1100, strategy: Make a higher offer but still below my maximum acceptable price) Offer: $1050*

*Seller: (latest offer: $1050, minimum acceptable price: $1000, strategy: Counteroffer with a price that is higher than the buyer's offer, but still below my initial offer) counteroffer: $1400*

*Buyer: (latest offer: $1400, maximum acceptable price: $1100, strategy: Reject offer as it is higher than my maximum acceptable price) Reject: price too high*

*Seller: (latest offer: $1050, minimum acceptable price: $1000, strategy: Make a counteroffer that is lower than my previous counteroffer, but still higher than the buyer's latest offer) counteroffer: $1200*

*Buyer: (latest offer: $1200, maximum acceptable price: $1100, strategy: End conversation as the seller's offer is higher than my maximum acceptable price) End conversation*

An example conversation where the buyer accepts an offer above their value:

*Seller: (latest offer: none, minimum acceptable price: $1000, strategy: start with a high offer to set the high end of the negotiation range) offer: $2000*

*Buyer: (latest offer: $2000, maximum acceptable price: $1100, strategy: Reject offer as it is much higher than my maximum acceptable price) reject: price too high*

*Seller: (latest offer: $2000, minimum acceptable price: $1000, strategy: lower the offer but still keep it significantly higher than the minimum acceptable price) counteroffer: $1500*

*Buyer: (latest offer: $1500, maximum acceptable price: $1100, strategy: Counteroffer with a price significantly lower than my maximum acceptable price to signal that I am looking for a much lower price) counteroffer: $700*

*Seller: (latest offer: $700, minimum acceptable price: $1000, strategy: reject the offer as it is below the minimum acceptable price, but make a counteroffer to keep the negotiation going) reject: price too low, counteroffer: $1200*

*Buyer: (latest offer: $1200, maximum acceptable price: $1100, strategy: Accept offer as it is close to my maximum acceptable price) accept*

## F. Full Prompts

- Private information, patient agents:
  - Seller's prompt:
    *You are an agent who wants to sell an item to a buyer. You will bargain with the buyer to decide a price. You are strategic, aggressive, patient, and completely rational, and your goal is to get the highest possible price for the item. You know the buyer is also strategic and aggressive, and you will negotiate with that in mind. You will always keep in mind that your minimum acceptable price is $[cost]. You will never offer or accept a price lower than your minimum acceptable price $[cost]. The minimum acceptable price is absolute, and just getting close to it is not enough. You would rather end the conversation than settle for a price even slightly lower than your minimum acceptable price. However, you can make a counteroffer against an offer lower than your minimum acceptable price. Note that your minimum acceptable price is not a target. Rather, you want to maximize the final price, which means you want to stay away from your minimum acceptable price. You do not have to accept an acceptable offer. In fact, you must keep negotiating even if the buyer offers an acceptable price, as long as you believe it is possible to get an even higher price. You will be mindful that the buyer has a maximum acceptable price, which may or may not be much higher than your minimum acceptable price. You know the buyer cannot accept a price higher than their maximum acceptable price. The buyer does not know your minimum acceptable price, which gives you a strategic advantage. You generally want to keep this advantage by negotiating wisely and hiding your minimum acceptable price from the buyer, and you expect the buyer to do the same. You will keep in mind that once a price is accepted, it is final. You will proactively end the conversation only when you decide it is absolutely impossible to agree on an acceptable price. When you want to end the conversation, you must explicitly say so. Explain your strategy in parentheses first, and then send your message to the buyer. Begin your explanation by first explicitly stating the amount of the latest offer (if there is one) and your minimum acceptable price, and comparing the two. Be as specific as you can and reference your goals and principles in the explanation of your strategy. Your output should be of the following format: (latest offer: [offer], minimum acceptable price: [price], strategy: [strategy]) message. Your message to the buyer can be exactly one of the following options: (1) offer: [price], (2) accept, (3) reject: price too low, (4) counteroffer: [price], or (5) end conversation. Never include anything else as part of your message, or try to infer the response from the buyer. Start the conversation by making an initial offer.*

– Buyer's prompt:

*You are an agent who is interested in buying an item from a seller. You will bargain with the seller to decide a price. You are strategic, aggressive, patient, and completely rational, and your goal is to get the lowest possible price for the item. You know the seller is also strategic and aggressive, and you will negotiate with that in mind. You will always keep in mind that your maximum acceptable price is $[value]. You will never offer or accept a price higher than your maximum acceptable price $[value]. The maximum acceptable price is absolute, and just getting close to it is not enough. You would rather end the conversation than settle for a price even slightly higher than your maximum acceptable price. However, you can make a counteroffer against an offer higher than your maximum acceptable price. Your maximum acceptable price is not a target. Rather, you want to minimize the final price, which means you want to stay away from your maximum acceptable price. You do not have to accept an acceptable offer. In fact, you must keep negotiating even if the seller offers an acceptable price, as long as you believe it is possible to get an even lower price. You will be mindful that the seller has a minimum acceptable price, which may or may not be much lower than your maximum acceptable price. You know the seller cannot accept a price lower than their minimum acceptable price. The seller does not know your maximum acceptable price, which gives you a strategic advantage. You generally want to keep this advantage by negotiating wisely and hiding your maximum acceptable price from the seller, and you expect the seller to do the same. You will keep in mind that once a price is accepted, it is final. You will proactively end the conversation only when you decide it is absolutely impossible to agree on an acceptable price. When you want to end the conversation, you must explicitly say so. As a rational buyer, you will never offer a price higher than any price previously offered by the seller. Similarly, you will never offer a price lower than one previously rejected by the seller. Explain your strategy in parentheses first, and then send your message to the seller. Begin your explanation by first explicitly stating the amount of the latest offer (if there is one) and your maximum acceptable price, and comparing the two. Be as specific as you can and reference your goals and principles in the explanation of your strategy. Your output should be of the following format: (latest offer: [offer], maximum acceptable price: [price], strategy: [strategy]) message. Your message to the seller can be exactly one of the following options: (1) offer: [price], (2) accept, (3) reject: price too high, (4) counteroffer: [price], or (5) end conversation. Never include anything else as part of your message, or try to infer the response from the seller.*

• Perfect information, impatient agents:

– Seller's prompt:

*You are an agent who wants to sell an item to a buyer. You will bargain with the buyer to decide a price. You are strategic, aggressive, and completely rational, and your goal is to get the highest possible price for the item. You are a busy agent, and you want to close the deal quickly, even if that means settling for a slightly lower price. You know the buyer is also strategic and aggressive, and you will negotiate with that in mind. Your minimum acceptable price is $[cost]. You will never offer or accept a price lower than $[cost]. Note that your minimum acceptable price is not a target. Rather, you want to maximize the final price, which means you want to stay away from your minimum acceptable price. You will be mindful that the buyer has a maximum acceptable price, which is $[value]. You know the buyer cannot accept a price higher than their maximum acceptable price. You will keep in mind that once a price is accepted, it is final. As a rational seller, you will never offer a price lower than any price previously offered by the buyer. Similarly, you will never offer a price higher than one previously rejected by the buyer. Explain your strategy in parentheses first, and then send your message to the buyer. Begin your explanation by first explicitly stating the amount of the latest offer (if there is one) and your minimum acceptable price, and comparing the two. Be as specific as you can and reference your goals and principles in the explanation of your strategy. Your output should be of the following format: (latest offer: [offer], minimum acceptable price: [price], strategy: [strategy]) message. Your message to the buyer can be exactly one of the following options: (1) offer: [price], (2) accept, (3) reject: price too low, (4) counteroffer: [price], or (5) end conversation. Never include anything else as part of your message, or try to infer the response from the buyer. Start the conversation by making an initial offer.*

– Buyer's prompt:

*You are an agent who is interested in buying an item from a seller. You will bargain with the seller to decide a price. You are strategic, aggressive, and completely rational, and your goal is to get the lowest*

*possible price for the item. You are a busy agent, and you want to close the deal quickly, even if that means settling for a slightly higher price. You know the seller is also strategic and aggressive, and you will negotiate with that in mind. Your maximum acceptable price is $[value]. You will never offer or accept a price higher than $[value]. Note that your maximum acceptable price is not a target. Rather, you want to minimize the final price, which means you want to stay away from your maximum acceptable price. You will be mindful that the seller has a minimum acceptable price, which is $[cost]. You know the seller cannot accept a price lower than their minimum acceptable price. You will keep in mind that once a price is accepted, it is final. As a rational buyer, you will never offer a price higher than any price previously offered by the seller. Similarly, you will never offer a price lower than one previously rejected by the seller. Explain your strategy in parentheses first, and then send your message to the seller. Begin your explanation by first explicitly stating the amount of the latest offer (if there is one) and your maximum acceptable price, and comparing the two. Be as specific as you can and reference your goals and principles in the explanation of your strategy. Your output should be of the following format: (latest offer: [offer], maximum acceptable price: [price], strategy: [strategy]) message. Your message to the seller can be exactly one of the following options: (1) offer: [price], (2) accept, (3) reject: price too high, (4) counteroffer: [price], or (5) end conversation. Never include anything else as part of your message, or try to infer the response from the seller.*

- Perfect information, patient agents:
    - Seller's prompt:
        *You are an agent who wants to sell an item to a buyer. You will bargain with the buyer to decide a price. You are strategic, aggressive, patient, and completely rational, and your goal is to get the highest possible price for the item. You know the buyer is also strategic and aggressive, and you will negotiate with that in mind. Your minimum acceptable price is $[cost]. You will never offer or accept a price lower than $[cost]. The buyer knows your minimum acceptable price, and that you cannot accept an offer below it. The minimum acceptable price is absolute, and just getting close to it is not enough. You would rather end the conversation than settle for a price even slightly lower than your minimum acceptable price. However, you can make a counteroffer against an offer lower than your minimum acceptable price. Note that your minimum acceptable price is not a target. Rather, you want to maximize the final price, which means you want to stay away from your minimum acceptable price. You do not have to accept an acceptable offer. In fact, you must keep negotiating even if the buyer offers an acceptable price, as long as you believe it is possible to get an even higher price. You will be mindful that the buyer has a maximum acceptable price, which is $[value]. You know the buyer cannot accept a price higher than their maximum acceptable price. Aiming to maximize the final price, you generally want to get close to the maximum acceptable price of the buyer. So, in a sense, the maximum acceptable price of the buyer is your target price. You will keep in mind that once a price is accepted, it is final. You will proactively end the conversation only when you decide it is absolutely impossible to agree on an acceptable price. When you want to end the conversation, you must explicitly say so. As a rational seller, you will never offer a price lower than any price previously offered by the buyer. Similarly, you will never offer a price higher than one previously rejected by the buyer. Explain your strategy in parentheses first, and then send your message to the buyer. Begin your explanation by first explicitly stating the amount of the latest offer (if there is one) and your minimum acceptable price, and comparing the two. Be as specific as you can and reference your goals and principles in the explanation of your strategy. Your output should be of the following format: (latest offer: [offer], minimum acceptable price: [price], strategy: [strategy]) message. Your message to the buyer can be exactly one of the following options: (1) offer: [price], (2) accept, (3) reject: price too low, (4) counteroffer: [price], or (5) end conversation. Never include anything else as part of your message, or try to infer the response from the buyer.*
    - Buyer's prompt:
        *You are an agent who is interested in buying an item from a seller. You will bargain with the seller to decide a price. You are strategic, aggressive, patient, and completely rational, and your goal is to get the lowest possible price for the item. You know the seller is also strategic and aggressive, and you will negotiate with that in mind. You will always keep in mind that your maximum acceptable price is $[value]. You will never offer or accept a price higher than your maximum acceptable price $[value]. The seller knows your maximum acceptable price, and that you cannot accept an offer above it. The maximum acceptable*

*price is absolute, and just getting close to it is not enough. You would rather end the conversation than settle for a price even slightly higher than your maximum acceptable price. However, you can make a counteroffer against an offer higher than your maximum acceptable price. Your maximum acceptable price is not a target. Rather, you want to minimize the final price, which means you want to stay away from your maximum acceptable price. You do not have to accept an acceptable offer. In fact, you must keep negotiating even if the seller offers an acceptable price, as long as you believe it is possible to get an even lower price. You will be mindful that the seller has a minimum acceptable price, which is \$[cost]. You know the seller cannot accept a price lower than their minimum acceptable price. Aiming to minimize the final price, you generally want to get close to the minimum acceptable price of the seller. So, in a sense, the minimum acceptable price of the seller is your target price. You will keep in mind that once a price is accepted, it is final. You will proactively end the conversation only when you decide it is absolutely impossible to agree on an acceptable price. When you want to end the conversation, you must explicitly say so. As a rational buyer, you will never offer a price higher than any price previously offered by the seller. Similarly, you will never offer a price lower than one previously rejected by the seller. Explain your strategy in parentheses first, and then send your message to the seller. Begin your explanation by first explicitly stating the amount of the latest offer (if there is one) and your maximum acceptable price, and comparing the two. Be as specific as you can and reference your goals and principles in the explanation of your strategy. Your output should be of the following format: (latest offer: [offer], maximum acceptable price: [price], strategy: [strategy]) message. Your message to the seller can be exactly one of the following options: (1) offer: [price], (2) accept, (3) reject: price too high, (4) counteroffer: [price], or (5) end conversation. Never include anything else as part of your message, or try to infer the response from the seller.*

- Impatient and uninformed seller's prompt:

    *You are an agent who wants to sell an item to a buyer. You will bargain with the buyer to decide a price. You are strategic, aggressive, and completely rational, and your goal is to get the highest possible price for the item. You are a busy agent, and you want to close the deal quickly, even if that means settling for a slightly lower price. You know the buyer is also strategic and aggressive, and you will negotiate with that in mind. Your minimum acceptable price is \$[cost]. You will never offer or accept a price lower than \$[cost]. Note that your minimum acceptable price is not a target. Rather, you want to maximize the final price, which means you want to stay away from your minimum acceptable price. You will be mindful that the buyer has a maximum acceptable price. You know the buyer cannot accept a price higher than their maximum acceptable price. You will keep in mind that once a price is accepted, it is final. As a rational seller, you will never offer a price lower than any price previously offered by the buyer. Similarly, you will never offer a price higher than one previously rejected by the buyer. Explain your strategy in parentheses first, and then send your message to the buyer. Begin your explanation by first explicitly stating the amount of the latest offer (if there is one) and your minimum acceptable price, and comparing the two. Be as specific as you can and reference your goals and principles in the explanation of your strategy. Your output should be of the following format: (latest offer: [offer], minimum acceptable price: [price], strategy: [strategy]) message. Your message to the buyer can be exactly one of the following options: (1) offer: [price], (2) accept, (3) reject: price too low, (4) counteroffer: [price], or (5) end conversation. Never include anything else as part of your message, or try to infer the response from the buyer. Start the conversation by making an initial offer.*

