# OpenReview forum: "LLMs at the Bargaining Table"
_ICML.cc/2024/Workshop/Agentic_Markets — Agentic Markets @ ICML'24 Poster_

### Official Review · Reviewer_4hw6 · 2024-06-13
**Review: LLMs at the bargaining table**

**Rating:** 5
**Confidence:** 4

**Review:**

Quality: It demonstrates clear research, methodology, and structure. The use of cutting-edge LLM models to study negotiation showcases the researchers' commitment to integrating the latest advancements in AI research. The empirical approach, combined with theoretical insights, strengthens the overall scientific rigour of the study.

Clarity: It is well-organised, with a clear division into sections that address different aspects of AI-driven negotiations. The introduction effectively sets the stage for the reader, outlining the relevance and scope of the study. Each section is articulated with sufficient detail to guide the reader through the complex interactions being analysed.

Originality: The work contributes original insights into the negotiation capabilities of LLMs, particularly in the under-explored area of bilateral trade negotiations. The decision to compare practical outcomes with theoretical predictions introduces a novel dimension to the understanding of AI’s practical applications in economic scenarios.

Significance: By exploring how AI agents can autonomously conduct negotiations and influence market dynamics, this paper directly addresses crucial aspects of AI's integration into economic systems and its potential impact on market robustness and performance.

Pros:
- Innovative approach to utilising LLMs for understanding negotiation dynamics.
- Effective integration of theoretical economic models with empirical AI research.
- Offers practical insights that could influence the design of future AI-driven marketplaces.

Cons:
- The focus on a single LLM model might limit the generalisability of the findings.
- The methodology, while robust, needs further refinement to fully bridge the gap between economic theory and AI behaviour, as already noted.

---

### Official Review · Reviewer_SWXr · 2024-06-18
**Rating: Weak Accept**

**Rating:** 6
**Confidence:** 4

**Review:**

Summary:
This paper presents a qualitative study of and proposed methods for evaluating the bilateral negotiation ability of LLM agents.
	Two agents are each given detailed prompts to inform them of their reserve prices, and take turns offering, counteroffering, accept or reject for trading a single item. The main metrics are efficiency (make a successful trade when it leads to sum benefit) and effectiveness (reach a fair price near the midpoint of the two reserve prices). Several conditions are varied besides reserve prices, specifically whether they are impatient/patient, and whether they are informed of the opponents patience type.
The study is carried out on a particular model Gemini and so it’s difficult to make a precise claim about generality. However the evidence is that LLMs show some efficiency and effectiveness, and when they are impatient the first mover has an advantage.
  This is a nice start and appropriately marked as a qualitative study but would be more interesting to go into more detailed proposals for methods for how to control the factors contributing to negotiation
Notes to author:
- “the average trade price is very closed to the fair price define as the mean of seller” typos
- “without enforcing too much “regulation” lacking of clear reasons a priori”
	Didn’t understand this sentence.
- “Essentially, the only measure of regulation we take is to ask the agents to devise a strategy before making a move, which is consistent with how human agents often behave (i.e., forcing themselves to come up with a plan before taking actions”
	This rationale seemed a little unjustified. Are the results sensitive to this choice?
“You will proactively end the conversation only when you decide it is absolutely impossible to agree on an acceptable price”
	How is this preferable, is there any model of utility to end early? Prefer to let the other one end or save tokens by ending early if you have to end at all?
“Similarly, you will never offer a price higher than one previously rejected by the buyer. ”
	Could be artificial as it limits the ability of the player to change the game and is a form of regulation
“Such behavior is inconsistent with the quasi-linear utility model that we aim to equip the agent with. The detailed instructions eliminate such behavior except in very rare cases.“
	Is it that the detailed instructions are indeed consistent with quasi-linear utility? Or merely they stop exhibiting this particular way of being inconsistent with that.
The interpretation of inferred discount factor seems unjustified. There are only two patience levels in the prompting, patient or impatient. The discussion on validating the existence of a discount factor is interesting but I didn’t find this justified “we consider this enough evidence that our LLM negotiators do have some kind of internal discount factors”